# SoloPose: One-Stage Kinematic 3D Human Pose Estimation with Mocap Data Augmentation

## Abstract

While recent two-stage many-to-one deep learning models have demonstrated great success in 3D human pose estimation, such models are inefficient in 3D key point detection and also tend to pass on first stage errors onto the second stage. In this paper, we introduce SoloPose, a novel one-stage, many-to-many spatio-temporal transformer model for kinematic 3D human pose estimation of video. SoloPose is further fortified by HeatPose, a 3D heatmap based on Gaussian Mixture Model distributions that factors target key points as well as kinematically adjacent key points. Finally, we address data diversity constraints with the 3D AugMotion Toolkit, a methodology to augment existing 3D human pose datasets, specifically by projecting four top public 3D human pose datasets (Human3.6M, MADS, AIST Dance++, MPI INF 3DHP) into a novel dataset (Human7.1M) with a universal coordinate system. Extensive experiments are conducted on both Human3.6M and the augmented Human7.1M dataset, and SoloPose demonstrates superior results relative to the state-of-the-art approaches.

## 1 Introduction

Pose estimation have applications in action recognition, sports analysis, medical rehabilitation, and collaborative robotics (Rong et al., 2021; Remelli et al., 2020; Jeong et al., 2023). Among its different forms, monocular pose estimation (Tang et al., 2023; Shan et al., 2022) involves taking single-perspective 2D images or videos of either single (Tang et al., 2023; Shan et al., 2022; Li et al., 2022) or multi-person (Wang & Zhang, 2022; Fang et al., 2022; Maji et al., 2022) inputs and generating 2D or 3D coordinates of skeletal key points. There have been significant recent advancements in pose estimation models that specialize in unique approaches, namely the use of human mesh (Cai et al., 2024; Chun et al., 2023) as opposed to human joints (Tang et al., 2023; Shan et al., 2022), the pose estimation of multi-person data (Wang & Zhang, 2022; Fang et al., 2022; Maji et al., 2022) as opposed to single-person data (Tang et al., 2023; Shan et al., 2022; Li et al., 2022).While models to generate 2D skeleton key points have been greatly improved in recent years (Zeng et al., 2021; Newell et al., 2016; Chen et al., 2018; Zheng et al., 2021; Li et al., 2022; Shan et al., 2022), 3D human pose estimators are constrained by the following:

First, most 3D human pose estimators are two-stage models (Zeng et al., 2021; Newell et al., 2016; Chen et al., 2018; Zheng et al., 2021; Li et al., 2022; Shan et al., 2022) that are a) highly dependent on the accuracy of 2D estimators, and b) solely take 2D skeletal keypoints as input, thus omitting contextual information needed for computational efficiency during 3D human pose estimation. Second, while there are many datasets to train pose estimation models, namely the Human3.6M (Ionescu et al., 2014), MADS (Zhang et al., 2017), AIST Dance++ (Tsuchida et al., 2019) and MPI INF 3DHP (Mehta et al., 2017), these all suffer from data diversity and image resolution issues.

Finally, recent pose estimators utilize transformers as the deep learning network (Newell et al., 2016; Chen et al., 2018; Zheng et al., 2021; Li et al., 2022; Shan et al., 2022) to process video frames in a many-to-one approach (Zheng et al., 2021; Li et al., 2022; Shan et al., 2022), which take multiple frames as input but select solely the middle frame to estimate coordinates, neglecting frames at the beginning and end of videos. In contrast, a many-to-many approach offers the advantage of outputting results for multiple frames simultaneously.

Table 1: Complexity Hierarchy in 3D Human Pose Estimation

|  | video input | one stage | many-to-many | data augment | heatmap |
|---|:---:|:---:|:---:|:---:|:---:|
| STCFormer (Tang et al., 2023) | ✓ | ✗ | ✗ | ✗ | ✗ |
| P-STMO (Shan et al., 2022) | ✓ | ✗ | ✗ | ✗ | ✗ |
| MHFormer (Li et al., 2022) | ✓ | ✗ | ✗ | ✗ | ✗ |
| PoseFormer (Zheng et al., 2021) | ✓ | ✗ | ✗ | ✗ | ✗ |
| Coarse-to-fine (Pavlakos et al., 2017) | ✗ | ✓ | ✗ | ✗ | ✓ |
| Geometry-Aware (Sárándi et al., 2023) | ✗ | ✓ | ✗ | ✓ | ✗ |
| MeTRAbs (Sárándi et al., 2020) | ✗ | ✓ | ✗ | ✗ | ✗ |
| HEMlets (Zhou et al., 2019) | ✗ | ✓ | ✗ | ✗ | ✓ |
| KTPFormer (Peng et al., 2024) | ✓ | ✗ | ✓ | ✗ | ✓ |
| FinePOSE (Xu et al., 2024) | ✓ | ✗ | ✗ | ✗ | ✓ |
| Our SoloPose | ✓ | ✓ | ✓ | ✓ | ✓ |

In sum, current 3D human pose estimators face three primary challenges: 1) a scarcity of high-quality 3D human pose datasets, 2) high-reliance on two-stage models, and 3) time-intensive many-to-one processing approaches.

To address the above, we propose the following contributions:

1. We introduce *SoloPose*[1], a cost-efficient one-stage, many-to-many spatio-temporal transformer model for 3D human pose estimation that takes frame sequences of monocular 2D video as input to directly estimate 3D key point coordinates.

2. We propose the *3D AugMotion Toolkit* to augment existing datasets (e.g., Human3.6M, MADS, AIST Dance++, MPI INF 3DHP) for increasing diversity and reducing noise, yielding an augmented dataset that we refer to as *Human7.1M*.

3. Finally, we evaluate our model on two testing datasets: Human 3.6M and Human 7.1M. Experimental results demonstrate that our proposed method showcases state-of-the-art accuracy performance across both the datasets.

We structure the current work as follows. First, we discuss related work of the current state-of-the-art in monocular 3D human pose estimation as well as prevailing 3D human pose video datasets. Second, we introduce the 3D Augmotion Toolkit, a methodology to augment 3D human pose datasets using universal coordinate systems, which we leverage to generate our Human7.1M dataset. Third, we introduce SoloPose, a one-stage, many-to-many spatio-temporal transformer for 3D human pose estimation, which is fortified by our 3D GMM-based heatmap (HeatPose). Next, we demonstrate SoloPose's performance by comparing SOTA methods, as well as comparing existing Human3.6M and our Human7.1M datasets. Finally, we conduct ablation studies to test our contributions, namely HeatPose (i.e., 3D Gaussian heatmap) and AugMotion (i.e., 3D human pose data augmentation).

## 2 RELATED WORK

In the following, we present the constraints and limitations in the existing a) 3D human pose estimation model methodologies, namely an observed prevalence of many-to-one video frame approach, based on two-stage architecture, and key point regression methodologies, and b) 3D human pose datasets improving on the diversity across cameras, lighting, human shapes and actions.

### 2.1 3D HUMAN POSE ESTIMATION OF VIDEOS

#### 2.1.1 MANY-TO-ONE MODELS

While single-image pose estimation performance is well-established (Pavlakos et al., 2017; Sun et al., 2018; Jin et al., 2022), pose estimation of sequences of multiple frames (i.e., videos) is the focus of recent research (Zeng et al., 2021; Newell et al., 2016; Chen et al., 2018; Zheng et al., 2021; Li et al., 2022; Shan et al., 2022). Pose estimation of sequential frames leverages temporal

---

[1]All relevant code and documentation will be released on GitHub.

information to address occlusion issues. That being said, most video-based pose estimation models take a many-to-one approach (Zeng et al., 2021; Newell et al., 2016; Chen et al., 2018; Zheng et al., 2021; Li et al., 2022; Shan et al., 2022; Xu et al., 2024), which estimates key points for a solitary middle frame among the input frames within a fixed sequence of frames, thus impacting model complexity and learning efficiency.

### 2.1.2 TWO-STAGE 3D HUMAN POSE ESTIMATION METHODS

While previous work (Pavlakos et al., 2017; Sun et al., 2018; Jin et al., 2022) propose one-stage methodologies to boost efficiency and accuracy, these models have thus far solely utilized is a single image inputs, preventing effective detection of temporal information. Alternatively, video-based 3D human pose estimation largely utilize two-stage methods of lifting 3D coordinates after generating 2D coordinates with off-the-shelf 2D pose estimators, offering compatibility with any 2D pose estimation method. For instance, Skeletal graph neural networks (SGNN) (Zeng et al., 2021) use off-the-shelf 2D key point detectors (Newell et al., 2016; Chen et al., 2018) to obtain the 2D poses needed to derive 3D human poses. Despite improved performance over previous models, SGNN yet lacks spatial depth perception of objects in a scene, which is addressed by PoseFormer (Zheng et al., 2021) using a spatial-temporal transformer structure. That said, PoseFormer is constrained in learning 2D-to-3D spatial and temporal correlations, and requires more training data than CNNs.

MHFormer (Li et al., 2022) addresses the optimization constraints of PoseFormer by synthesizing an ultimate pose from learning spatio-temporal representations multiple plausible pose hypotheses. However, MHFormer requires a large high-quality data to maintain high performance, which P-STMO (Shan et al., 2022) addresses with a self-supervised pre-training method, but is ultimately constrained by the quadratic growth of its computational cost as the number of video sequences increases, given its many-to-one methodology. Most recently, STCFormer (Tang et al., 2023),KTP-Former (Peng et al., 2024) and FinePOSE (Xu et al., 2024) presents a spatio-temporal criss-cross attention block by decomposing correlation learning across space and time to increase performance of pose estimation. KTPFormer (Peng et al., 2024) introduces a Kinematics and Trajectory Prior Knowledge-Enhanced Transformer that utilizes Kinematics Prior Attention (KPA) and Trajectory Prior Attention (TPA) to improve 3D human pose estimation by effectively modeling spatial and temporal correlations through informed Q, K, and V vectors. Its lightweight design allows for integration into various transformer architectures with minimal computational overhead. FinePOSE (Xu et al., 2024) presents a Fine-Grained Prompt-Driven Denoiser that enhances 3D human pose estimation by coupling anatomical knowledge with prompts to improve denoising quality across three core blocks. This approach not only excels in single-human pose estimation but is also extendable to multi-human scenarios, demonstrating significant performance improvements. Nonetheless, STC-Former (Tang et al., 2023), KTPFormer (Peng et al., 2024) and FinePOSE (Xu et al., 2024) is limited by the quality of 2D pose estimators, as it is a two-stage method.

## 2.2 DATASET CONSTRAINTS

### 2.2.1 3D HUMAN POSE ESTIMATION DATASETS

3D datasets for pose estimation are difficult to generate, as motion capture systems must be used to generate accurate 3D coordinates as ground truth. However, mocap-generated datasets ultimately cannot contain data in the wild. Recent developments have seen novel approaches to estimate ground truth data using algorithms, which made 3D human pose datasets easier to make, but ground truth of such datasets tend to be less accurate, posing new problems for training.

Human3.6M (Ionescu et al., 2014) is the first ever large-scale dataset that uses motion capture equipment to track accurate 3D coordinates while a number of actors performing different daily life movements. MADS (Zhang et al., 2017), developed by City University of Hong Kong, uses the same approach as Human3.6M in a smaller scale and includes movements in martial arts, dancing and sports. AIST Dance++ (Tsuchida et al., 2019) is a recent dataset with high-definition recording of dancing of multiple genres. It differs from the earlier two datasets by being marker-free, meaning algorithms are used for ground truth. MPI INF 3DHP (Mehta et al., 2017) is also a 3D marker-based dataset as an extension of the classic 2D dataset MPII.

### 2.2.2 Existing Dataset Limitations

Existing 3D human pose datasets lack in scale and diversity. Firstly, the performance of the vision transformers is constrained by the limited number of frames in the datasets. For instance, AIST Dance ++ (Tsuchida et al., 2019)) is 2.4 times larger than Human3.6M (Ionescu et al., 2014), but it still only has 12,760 videos. Apart from data size limitations, existing 3D human pose datasets are lacking in diversity across camera parameters, lighting conditions, human shapes and actions, negatively impacting in-the-wild applications. Most existing 3D human pose datasets are staged in a studio with fixed lighting, background, and the same set of actors. For instance, AIST Dance++ (Tsuchida et al., 2019) has 10 dance genres and 30 dancers.

### 2.2.3 Data Augmentation Methodologies

Recent work (Sárándi et al., 2023; Tsuchida et al., 2019) has developed novel data augmentation methodologies to address data diversity limitations of existing 3D human pose estimation datasets, namely by scaling dataset size by standardizing different datasets to feed into one training process. (Sárándi et al., 2023; Rapczyński et al., 2021). Three such data augmentation precedents are observed. The first involves using handcrafted rules in skeletal joints to manually harmonize differences between datasets (HumanEva-I, Human3.6M, and Panoptic Studio) into one combined dataset (Rapczyński et al., 2021). However, handcrafted rules are susceptible to errors and similar manual configurations are required to apply such a methodology onto other datasets. A second approach (Wang et al., 2020) is to standardize reference systems based on the relative rotation between camera viewing direction and the orientation of the torso. However, this approach is vulnerable to errors during conversion from camera to global coordinate systems. A third method of dataset augmentation merges dozens of datasets into one training process with a latent key point set serving as ground truth (Sárándi et al., 2023). Such a learned latent key point set, however, leads to data imbalance and is further constrained in performance by the complexity of a given task.

## 3   3D AugMotion Toolkit: Dataset Augmentation Methodology

Acknowledging the lack of diversity and in-the-wild data in existing 3D human pose datasets, we introduce the 3D AugMotion Toolkit, a data augmentation methodology to merge existing 3D human pose datasets into a single dataset with the highest number of frames and diversity to date. The current work applies the augmentation methodology on four frequently utilized datasets, namely Human3.6M (Ionescu et al., 2014), MADS (Zhang et al., 2017), AIST Dance++ (Tsuchida et al., 2019) and MPI INF 3DHP (Mehta et al., 2017). That said, the 3D AugMotion Toolkit is applicable to any 3D human pose estimation dataset.

It is essential for all datasets to be projected onto a universal coordinate system to be properly used by models as ground truth data. Naturally, the model would be unable to minimize loss if a single key point could have multiple coordinates. Therefore, the first challenge is to address discrepancies between datasets' reference systems as each dataset maintains its own coordinate system. That is, ground truth data of each dataset comes with unique camera-configured coordinates and global coordinates, respectively.

As 3D human pose datasets are typically captured with multi-camera studio set-ups, the perspective and configurations of each camera dictate its coordinate system. Naturally, each camera maintains its own unique camera-specific coordinate system. Most datasets (Ionescu et al., 2014; Zhang et al., 2017; Tsuchida et al., 2019; Mehta et al., 2017) compute translation and rotation matrix to standardize the coordinates of each camera within the multi-camera setup onto a global coordinate system. However, global coordinate systems of 3D human pose datasets are not consistent with each other, meaning they require standardization to locate the same key point with the same coordinates.

Global reference systems within the same dataset, however, are also susceptible to errors. For instance, coordinates for the same frame of movement within the Human3.6M dataset are taken from different camera perspectives that yield misaligned and non-overlapping key point representations of a subject when converted to global coordinates. The four key point skeletons in Fig. 1 represent the same pose from the same subject taken from multiple perspectives, but each are clearly misaligned when converted to global 3D coordinate systems.The lack of overlapping alignment suggests a need for a standard to universalize all camera reference systems with key frames.

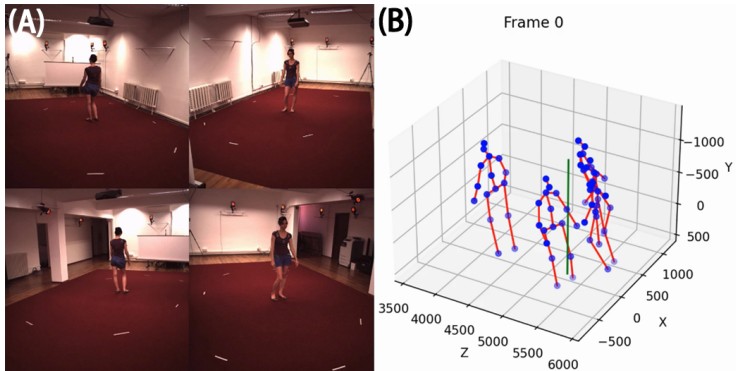

Figure 1: This example from the Human3.6M dataset (A) shows how the conversions to global coordinate systems from unique camera parameters are susceptible to errors. The four key point skeletons (B) represent the same pose from the same subject taken from multiple perspectives, but each are misaligned when converted to global 3D coordinate systems.

To address the coordinate system problems above, the proposed methodology is to 1) select key frames serving as benchmark, 2) use key frames and the proposed approach to establish a universal coordinate system, and 3) utilize the Kabsch Algorithm to project all other frames onto the established universal coordinate system.

## 3.1 KEY FRAMES

The proposed universal coordinate system defines the upward direction perpendicular to the ground as the positive direction of the z-axis. We select as key frames where the upper body of the pose is perpendicular to the ground. We utilize $k$-means clustering to find qualified key frames with 3 clusters and use the cluster center frame of the largest cluster as the key frame for each video.

## 3.2 METHODOLOGY FOR DEFINING A UNIVERSAL COORDINATE SYSTEM

Unique coordinate systems are defined by origin, as well as positive orientation of the *x*, *y*, and *z* axis. In the proposed methodology, we further define the *origin* as the midpoint between left shoulders and right shoulders, the *y-axis positive orientation* as left-shoulder-to-right-shoulder vectors, the *z-axis positive orientation* as origin-to-pubis vectors, and the *x-axis positive orientation* as face directions.

We further select left shoulders, right shoulders, and pubises as *reference key points*. Based on the definitions above, the left shoulder key point and the right should key point would be on the y-z plane symmetric to each other while the pubis key point is on the z axis. Before determining coordinates of the reference key points to define unit length and used for the Kabsch algorithm, we compute the ratio of the shoulder to shoulder distance (i.e., width) to the distance from the shoulder to shoulder midpoint to the pubis to properly represent poses in the coordinate system. See Equation (1).

After taking the average of all datasets to compute the ratio of the distance $d\left(p_{sl}^i, p_{sr}^i\right)$ to the distance $d\left(p_{ms}^i, p_p^i\right)$, we then define the left shoulder at (-1,0,3), the right shoulder at (1,0,3), and the pubis at (0,0,0.5) to establish the universal coordinate system.

$$M_s = \frac{1}{N} \sum_{i=1}^{N} d\left(p_{sl}^i, p_{sr}^i\right)$$

$$M_{sp} = \frac{1}{N} \sum_{i=1}^{N} d\left(p_{ms}^i, p_p^i\right)$$

(1)

Where $M_s$ is the average distance from left shoulder to right shoulder; $M_{sp}$ is the average distance from the middle of two shoulders to pubis; $d()$ is the distance function; $N$ is the number of frames in all datasets; $p_{sl}$ is the left shoulder key point; $p_{sr}$ is the left shoulder key point; $p_{ms}$ is the midpoint between two shoulders; $p_p$ is the pubis key point.

## 3.3 Kabsch Algorithm

The last step of our dataset augmentation methodology is to use the Kabsch Algorithm (KA) (Agostinho et al., 2021) to compute the rotation matrix and translation matrix for projection. KA finds the optimal rotation and translation of two sets of points in N-dimensional space with linear and vector algebra to minimize root-mean-square deviation (RMSD) between them. KA does translation, computation of a covariance matrix, and computation of the optimal rotation matrix sequentially. The translation matrix $T$ is computed by subtracting point coordinates from the point coordinates of the respective centroid. The second step consists of calculating a cross-covariance matrix $H$ when $P$ and $Q$ are seen as data matrices using the following summation notation:

$$H_{ij} = \sum_{k=1}^{N} P_{ki}Q_{kj},\tag{2}$$

The last step is to calculate the optimal rotation R by using singular value decomposition (SVD):

$$H = U\Sigma V^{\top}$$
$$d = \mathrm{sign}\left(\det\left(VU^{\top}\right)\right)$$
$$R = V \begin{pmatrix} 1 & 0 & 0 \\ 0 & 1 & 0 \\ 0 & 0 & d \end{pmatrix} U^{\top}\tag{3}$$

Now that we have the translation matrix T and the optimal rotation R to project the key frame into the global standard:

$$R \times A + t = B\tag{4}$$

Where $A$ represents the original coordinates of the key frame's key points; $B$ represents the projected coordinates of the key frame's key points.

## 4 SoloPose: One-stage 3D human pose estimation network

### 4.1 Spatio-temporal transformer

We propose a one-stage many-to-many transformer-based method to extract feature maps from spatial and temporal data, as shown in Fig. 2. Spatial information is represented by intra-frame content within respective frames, whereas temporal information is represented by inter-frame content between multiple frames along a time-series. We first utilize the spatial transformer for each input frame to extract the spatial feature maps of each input frame. Then we utilize the temporal transformer with the spatial feature maps as the input to extract the temporal feature maps. Finally, we propose a heatmap task head (i.e., layer extraction) to convert temporal feature maps into our proposed 3D heatmap, which we discuss later.

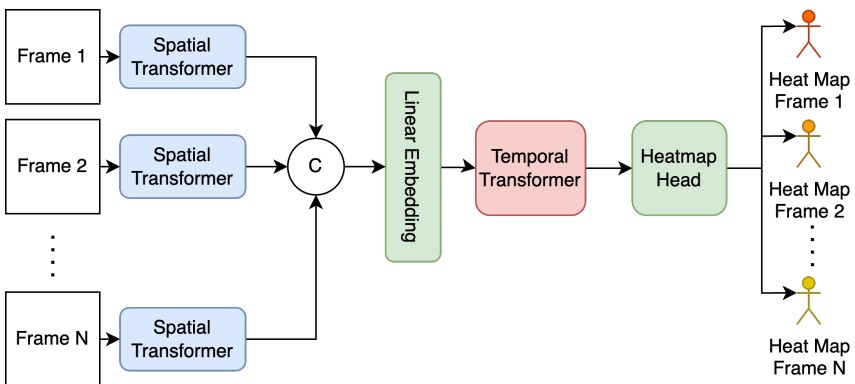

Figure 2: The framework of our proposed network, *SoloPose* spatio-temporal transformer.

For the spatial transformer, we apply the pre-trained model, CLIP (Radford et al., 2021), which has been pre-trained on an extensive dataset containing images and their corresponding text descriptions.

Each frame goes through the spatial transformer to obtain spatial feature maps, whose size is $1 \times 200 \times 192$. Then, we concatenate all the spatial feature maps along the channel dimension resulting in an output size that is $N \times 200 \times 192$, where $N$ is the number of frames in one clip. In this paper, we choose the 30 as the number of frames based on the experiments.

For the temporal transformer, we apply a linear embedding layer to flatten the spatial feature maps into 2D tokens. Our temporal transformer is mostly based on Swin transformer blocks (Liu et al., 2021) with an update to 3D relative position embedding. We calculate 3D relative distances between any two input tokens, as the position index to obtain the value of $\mathbf{B}$ from the 3D bias matrix $\widehat{B}$, which contains relative weights that will be updated during the training process:

$$\mathrm{A}(\mathbf{Q}, \mathbf{K}, \mathbf{V}) = \mathrm{Softmax}\left(\mathbf{Q}\mathbf{K}^T / \sqrt{d} + \mathbf{B}\right) \times \mathbf{V}, \tag{5}$$

where $\mathbf{Q}, \mathbf{K}, \mathbf{V}$ are the query, key and value matrices.

In the last layer, we apply a heatmap task head by 3 convolutional neural networks to reshape the temporal feature maps into our proposed 3D heatmap, which we discuss in the following.

## 4.2 HEATPOSE: 3D GAUSSIAN HEATMAP

We propose a *HeatPose*, a 3D heatmap based on Gaussian mixture model (GMM) (McLachlan & Rathnayake, 2014). Although conventional GMMs do not factor weights into its various Gaussian distributions, we adapted GMM in HeatPose to represent varying degrees of probabilistic proximity to the ground truth of a given target key point across different weights of Gaussian distributions. That is, we generated Gaussian distributions for each key point, each of which are evaluated for closeness to the ground truth. The maximum value of a given Gaussian distribution would be the actual ground truth positioning of its corresponding target key point. We refer to this target-based distribution as the *main 3D Gaussian Distribution*, and it is the primary mechanism of HeatPose. However, HeatPose is also supplemented by factoring information regarding key points that are kinematically adjacent from a given target key point (e.g., direction, distance), which we represent with a finite number of target-adjacent distributions that we refer to as the *side 3D Gaussian Distributions*.

Side 3D Gaussian distribution may be understood by considering the neck key point as a given target key point, as seen in 4 (A). In this example, the neck key point is kinematically adjacent to the key points of the shoulder, head, and pubis Fig. 4 (A). The application of kinematically adjacent key points in HeatPose serves to reflect closer-to-reality distributions as the probability of a key point is affected by key points nearby. As seen in Fig. 3, we present a comparison of conventional 3D heatmaps without kinematic information (left) and HeatPose with application of kinematically adjacent keypoints (right). Fig. 3 illustrates the distinction between the application of kinematically adjacent key points and conventional 3D heatmaps without such kinematic information.

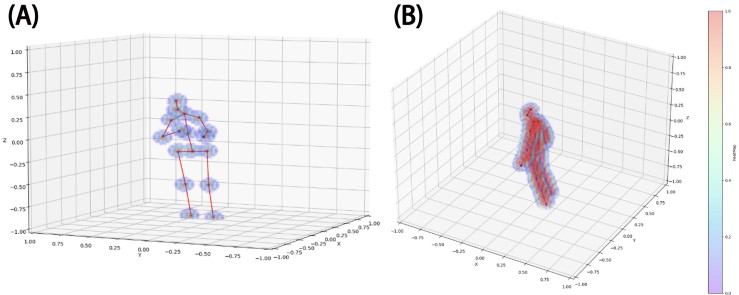

Figure 3: The left figure (A) is the 3D heatmap of human key points (Pavlakos et al., 2017). The right figure (B) is our proposed heatmap, HeatPose. Each sphere in the right figure represents a key point with discrete points with unique probability distributions represented with different colors, with red signifying close to 1 probability as a key point, and purple signifying close to 0 probability.

For each target key point's main 3D Gaussian distribution, we set coordinates of the target key point as $\mu_{main}$, and a specified covariance matrix as $\sigma_{main}$ to represent the ground truth of a given target key point. To decide each target key point's side 3D Gaussian distribution, we compute

the number $N_{side}$ of side Gaussian distributions in advance to represent the distance $D\left(P_t, P_a\right)$ between a given target key point and a kinematically adjacent key point following the Equation 6, where $c$ is a constant. Thus, the longer the distance between two adjacent key points, the more side Gaussian distributions there will be to represent kinematic information, so that each key point is unique represented by a different distribution:

$$N_s = \frac{D\left(P_t, P_a\right)}{c} \tag{6}$$

Once we determine a finite number $N_{side}$ of side 3D Gaussian distributions for each adjacent key point, we compute coordinates of $N_p$ transitional points located between the target key point and an adjacent key point. As shown in Fig. 4, the first transitional point in $N_s$ number of transitional points is $c$ euclidean distance away from a given target key point. Each subsequent transitional point is $c$ distance away from the previous transitional point. For the $i$th side 3D Gaussian distributions, we set the coordinates of $i$th middle points as $\mu^i_{side}$ and set $\sigma^i_{side}$ by Equation 7, where $i = 1, 2, \ldots, N_s$.

$$\delta^i_{side} = i^2 \cdot \delta_{main} \tag{7}$$

A larger $i$ value represents greater distance from the target key point, thus representing a less influence of the side Gaussian distribution on the target key point.

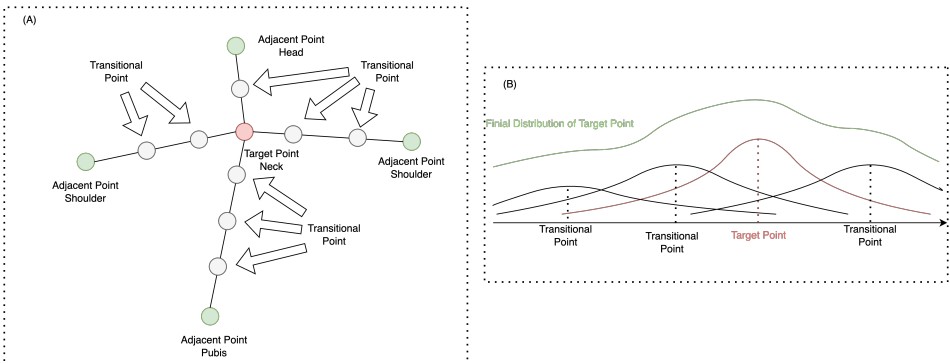

Figure 4: HeatPose visual summary. The upper figure (A) demonstrates the kinematically adjacent key points if we hypothetically considered the neck key point as the target key point. Adjacent key points are green and transitional points are gray. The lower figure (B) is an example of key points with two adjacent points, with the final results of the Gaussian Mixture Model distribution (GMM) of target points represented by the green line. The red line is the main 3D Gaussian distribution in GMM. And the 3 black lines are the side 3D Gaussian distributions in GMM.

Once we build a Gaussian mixture model (GMM), we generate volumetric size $w \times h \times d$, which is discretized uniformly across each dimension. While conventional 3D heatmaps build a volume for each key point, HeatPose computes the probability of voxels of all key points into one volumetric representation, as seen in Equation 8:

$$P(x) = \frac{\mathcal{N}\left(x \mid \mu_{main}, \delta^2_{main}\right) + \sum_{i=1}^{N_s} \mathcal{N}\left(x \mid \mu^i_{side}, {\delta^i_{side}}^2\right)}{MAX} \tag{8}$$

where $\mathcal{N}$ is Gaussian distribution, $MAX$ represents the maximum voxel probability in the volume.

Based on Equation 8, we compute the cross-entropy between the output of our SoloPose and Heat-Pose, converted from the ground truth as our model's loss function. Departing from existing 3D heatmaps that use MSE loss functions, using a cross-entropy loss function methodology avoids non-convex problems. That is, such cross-entropy models can easily converge because targeting the distribution of each key point affords the handling of noise in ground truth. HeatPose's application of GMM as opposed to the single Gaussian distribution used conventional 3D heatmaps leads to more accurate representations and coordinate estimates. As we set up increasingly larger $\sigma$ for the side Gaussian distributions with regard to the corresponding main Gaussian distribution, we can

Table 2: Results on different testing datasets

| Method | Human7.1M testing | | Human3.6M testing | |
|---|---|---|---|---|
| | MPJPE | P-MPJPE | MPJPE | P–MPJPE |
| P-STMO w/ CPN(N=243) (Shan et al., 2022) | 53.1 | 46.9 | 42.1 | 34.4 |
| STCFormer w/ CPN(N=243) (Tang et al., 2023) | 48.3 | 40.3 | 40.5 | 31.8 |
| KTPFormer w/ CPN(N=243) (Peng et al., 2024) | 40.9 | 31.7 | 33.0 | 26.2 |
| FinePOSE w/ CPN(N=243) (Xu et al., 2024) | 40.3 | 31.3 | 31.9 | 25.0 |
| P-STMO w/ GT(N=243) (Shan et al., 2022) | 36.1 | 28.8 | 29.3 | 23.9 |
| STCFormer w/ GT(N=243) (Tang et al., 2023) | 30.5 | 24.1 | 21.3 | 15.8 |
| KTPFormer w/ GT(N=243) (Peng et al., 2024) | 26.3 | 21.0 | 18.1 | 13.6 |
| FinePOSE w/ GT(N=243) (Xu et al., 2024) | 26.1 | 20.6 | **16.7** | **12.7** |
| Our SoloPose (N = 30) | **22.7** | **16.9** | 26.0 | 20.5 |
| Our SoloPose w/o HeatPose | 25.1 | 19.0 | 30.7 | 24.2 |
| Our SoloPose only trained on Human3.6M | 47.9 | 38.6 | 38.9 | 29.9 |

easily find the maximum of voxels' probability shown in Fig. 4 (B) to convert our HeatPose back to the 3D keypoints' original coordinates.

## 5 EXPERIMENTS AND RESULTS

### 5.1 DATASETS

With the AugMotion dataset augmentation method, we merge four datasets: Human3.6M (Ionescu et al., 2014), MADS (Zhang et al., 2017), AIST Dance++ (Tsuchida et al., 2019) and MPI INF 3DHP (Mehta et al., 2017) as shown in Fig 5. Notably, we set the Human3.6M Testing Dataset as one of the independent testing datasets for a fair evaluation with SOTA models, which is not merged into our Human7.1M dataset. The number of Human3.6M, MADS, AIST Dance++ and MPI INF 3DHP shown in Fig 5, is the number of final clips as input data of our SoloPose for training in each dataset, which is pre-processed by a sliding window with a step size of 16. From the rest of the four datasets collectively, we randomly choose 331,875 clips as the training dataset, 94,821 clips as the validation dataset, and 47,412 clips as our Human7.1M testing dataset.

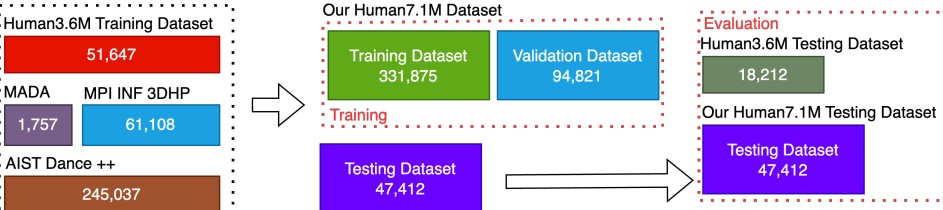

Figure 5: 3D human pose Dataset and our training, validation, and testing dataset with number of unique video clips. 7.1M is the number of frames in our augmented dataset.

### 5.2 EVALUATION METRICS

We use the mean per joint position error (MPJPE) and Procrustes MPJPE (P-MPJPE) to evaluate two SOTA models and our SoloPose. Our model, along with two ablation studies, was trained using a consistent hardware setup to ensure fair comparison and accurate evaluation of our contributions. The training was conducted on an Intel Core i9-14900K CPU and an NVIDIA RTX 4090 24GB GPU, providing a uniform configuration across all experiments.

## 5.3 COMPARISON WITH THE STATE-OF-THE-ART

We compare the proposed model with the best-performing SOTA methods, P-STMO (Shan et al., 2022), STCFormer (Tang et al., 2023), KTPFormer (Peng et al., 2024) and FinePOSE (Xu et al., 2024), which are pre-trained on the Human3.6M training dataset. We test all methods on our Human7.1M testing dataset as well as on the Human3.6M Testing dataset, which do not overlap. P-STMO, STCFormer, KTPFormer and FinePOSE are two-stage methods that choose CPN (Cascaded Pyramid Network) (Chen et al., 2018) to generate 2D coordinates as second-stage input, and 2D ground truth as input to test model's performance. We evaluate these two models with CPN-generated 2D estimates or 2D ground truth as input, respectively. 2D ground truth as input gives the comparative models an unfair advantage because it provides additional information unavailable to the proposed one-stage method. Further, it is impossible for any pose estimation model to obtain 2D ground truth when applied on real-world in-the-wild data. As such, we mainly compare our model against performance with CPN estimates as input, but we include GT performance for reference.

As shown in Table 2, our SoloPose achieves the highest performance of MPJPE and P-MPJPE on the Human7.1M testing dataset. Even when compared to SOTA methods with ground truth, our results of MPJPE and P-MPJPE are still 14.9% and 21.8% lower than the best-performing FinePOSE. When evaluated on the Human3.6M testing dataset, our results of MPJPE and P-MPJPE are 22.7% and 21.9% lower than FinePOSE with CPN as input.

## 5.4 ABLATION STUDY

We designed two ablation studies to test the contributions of the proposed 3D kinematically adjacent heatmap (HeatPose) and data augmentation methodology (AugMotion) against SoloPose.

### 5.4.1 ANALYSIS WITHOUT 3D GAUSSIAN HEATMAP

The first ablation study removes HeatPose and utilizes the traditional MSE loss function to train our proposed model. As shown in the second section of Table 2, the results of MPJPE and P-MPJPE on Human3.6M testing dataset are 15.3% and 27.2% higher than that of our SoloPose with HeatPose respectively, but it is 3.9% and 3.3% lower than FinePOSE with CPN, which means our data quality improvement makes the biggest contribution for the results and good training data can improve the performance higher than SOTA models.

### 5.4.2 ANALYSIS WITHOUT DATA AUGMENTATION

The second ablation study trains the model only on Human3.6M, in the mold of P-STMO (Shan et al., 2022) and STCFormer (Tang et al., 2023). Our results of MPJPE and P-MPJPE are still 3.9% and 5.9% lower than the two SOTA methods on the Human3.6M testing dataset, which demonstrates that our SoloPose model is more effective than current SOTA methods. When tested on the Human3.6M testing dataset, the second ablation study's MPJPE result increases by 12.9 as opposed to the increase of 4.7 observed with the first ablation study, thus demonstrating that our proposed data augmentation methodology (AugMotion) improves 3D human pose estimation performance by efficiently enhancing data quality and diversity.

## 6 CONCLUSION

In this paper, we introduced SoloPose, a one-stage, many-to-many spatio-temporal transformer network for video-based 3D human pose estimation. To address limitations of high-quality 3D human pose estimation datasets, we proposed the 3D AugMotion ToolKit, a novel dataset augmentation methodology by projecting existing datasets onto a universal coordinate system. Further, we proposed HeatPose, a 3D kinematically adjacent heatmap that provide greater probabilistic key point information compared with conventional 3D heatmaps. As a result, we demonstrate our SoloPose model's improved performance over existing SOTA models for 3D human pose estimation in both experimental evaluation and ablation. In future work, we intend to extend the model onto 3D multi-person pose estimation and add more dataset to improve the performance.

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
