# OpenReview forum: "SoloPose: One-Stage Kinematic 3D Human Pose Estimation with Mocap Data Augmentation"
_ICLR.cc/2025/Conference — Submitted to ICLR 2025_

### Official Review · Reviewer_pJ78 · 2024-10-21

**Soundness:** 2
**Presentation:** 1
**Contribution:** 2
**Rating:** 3
**Confidence:** 4

**Summary:**

The paper proposes a single-stage approach for 3D HPE. It uses the frames of videos directly as input sequences and outputs also a sequence of results for all input frames. The architecture is similar to existing 2D-to-3D uplifting pipelines consisting of a spatial and temporal Transformer part. The method uses special 3D-heatmaps as the target encoding. The heatmaps are created by using a gaussian mixture model and thex incorporate not only the keypoints one wants to detect but also intermediate points. Further, the authors combine multiple datasets and train jointly on them. In their evaluations they show that their method trained on all datasets achieves superior performance regarding the Human3.6m test set than SOTA 2D-to3D methods based on CPN 2D detections and trained only on Human3.6m.

**Strengths:**

The authors propose a method to unify multiple 3D HPE datasets. The model that they introduce achieves SOTA results on the Human3.6m dataset. The proposed model is a single stage pipeline and outputs the results for multiple frames at once. The authors introduce a novel method for 3D heatmaps. They can prove in the ablation study that this improves their model.

**Weaknesses:**

The writing and presentation quality is poor. There are many errors in the writing (incomplete sentences, missing words, etc.). Further, many explanations are vague and unclear to me, I will specify the problems the following.

The introduction section is not the related work section. An enormous amount of citations makes it hard to get the real motivation and to follow the flow of the text. In contrast, the authors state that they present the constraints and limitations of current work in the related work section. However, this is not the purpose of the related work section. The related work section is hard to read if the reader does not know the mentioned papers. Clearer writing and more concise explanations are required.

Table 1 does not help. One stage methods are not worse per se. The same holds for data augmenation and heatmap methods.
The authors state that a primaty challenge of 3D HPE is the high reliance on two-stage methods. This is not a problem at all. Why should it be?

The authors name as another challenge the time consumption of current approaches. However, throughout their paper, they do not provide any runtime analysis of their own model and no runtime comparison of their model to other models. This is a main weakness, since the authors use this as a main motivation.

The authors propose a 3D AugMotion toolkit with the purpose to merge 3D HPE datasets. This is NOT data augmentation. Data augmentation means to make slight changes to already existing data in order to "artificially" increase the dataset size. In this case, the dataset size is increased with real data from other datasets.

The authors do not mention one benefit of 2D to 3D uplifting models. They can also be trained on large MoCap datasets without available image data like AMASS. This is a benefit compared to the method proposed in this paper.

I do not understand the explanation of the definition of the universal coordinate system at all. Why are key frames needed and why are the key frames with upper body perpendicular to the ground (lines 242++)? What does k means clustering cluster here? Are there frames where the upper body is exactly perpendicular at all?

In lines 258++, the authors set exact positions for the left and right shoulder and the pubis. How is that possible for more than one person? All humans have different sizes and it is impossible to define these keypoints for two distinct humans. What is $M_s$ and $M_{sp}$ needed for? I can not follow here. What is the benefit of this at all? $p_{sr}$ should be the right shoulder keypoint, not the left a second time (line 269).

How do the authors deal with different keypoint definitions, meaning that the position of e.g. the left and right hip joint might differ in other datasets, relative to the body surface/volume. Further, different datasets contain different keypoints. How is this managed?

The application of the Kabsch Algorithm is explained very poorly. I can not understand how it is applied. Where do P and Q come from? What is the input and the output of the algorithm? And what does it help for the final goal?

The SoloPose approach introduced by the authors operates on frame sequences and outputs sequences of 3D heatmaps. A sequence of frames is a lot more memory consumptive than a sequence of 2D poses. Therefore, the authors are only able to set their sequence length to 30, which is comparably low to 2D to 3D uplifting pipelines with sequence leghts of up to 350. The authors do not provide any insights about the sequnce length they use and about memory consumption. They further do not explain how many transformer blocks they use in total.

The authors state that they incorporate a 3D relative positional embedding. I do not understand how they can calculate the 3D relative distances between tokens and what the 3D bias matrix B should be, and anything else in lines 330-336.

The description of using "3 convolutional neural networks" as the last layer is bad phrasing. Is it 3 CNN layers? Or complete networks? How are convolutions applied to the 1D tokens? Is it 1D convolutions?

I do not get how the final 3D heatmap is created. The explanation should provide all steps that the reader knows how to implement it in the end. Figure 3 contains images with too small numbers and is pixelated. Figure 4 presents upper and lower figure but the figures are side by side. If I understand correctly, there is one 3D heatmap for all keypoints, not one per keypoint. However, how do the authors extract the needed keypoints in the end? How is the postprocessing from network output to final coordinate? How is the discretization achieved? It is not explained at all. Equation 6 seems like it would be possible to introduce more transition points if a person is bigger than another one. Basically, the explanations of all section 4.2 are insufficient.

Table 2 is hard to interpret. A column indicating the usage of GT or not would help.

Why is the validation set of the proposed Human7.1M dataset larger than the test set? Further, the authors do not state that they aim to make it available for other researchers. Moreover, it seems to be targeted only to their setup, since it contains of clips with a sliding window and not frames with annotations or full videos.

The comparison of the proposed method with other methods is unfair. It is not fair to compare a model that is trained on Human7.1m with models that are solely trained on Human3.6m, especially as the validation dataset contains motions like dancing and the training dataset Human3.6m only everyday activities. The score of SoloPose trained only on Human3.6m is a lot worse, which proves this. Further, the proposed model achieves better results than the other methods with CPN as the estimator for 2D poses, but worse results than with using 2D GT poses. To achieve a fair comparison, GT poses are too good, but CPN is not SOTA any more and with using a better 2D estimator, the performance can be increased a lot.

When indicating improvements of the method, the authors should mention the unit (lines 524++)

**Questions:**

Most questions are described in the weaknesses section. Most important are the explanations of the 3D heatmap creation and the approach to unify the different datasets. How are these exactly achieved? What are the steps that are executed and why? The paper is very confusing for me and I could not follow.

How does the proposed approach perform in comparison to other SOTA models regarding runtime and memory consumption? Is it more efficient?

How would other SOTA models perform when trained on Human7.1m? Why do you not plan to make it available or do you?

More explanations of details from the architecture would help, e.g. what are the input and output sizes of the different transformer blocks?

See weaknesses section for more.

**Details Of Ethics Concerns:**

The paper deals with human data, but the used datasets are publicly available and commonly used for years.

---

### Official Review · Reviewer_yUKf · 2024-10-29

**Soundness:** 2
**Presentation:** 2
**Contribution:** 2
**Rating:** 5
**Confidence:** 3

**Summary:**

The paper presents a one-stage, many-to-many spatiotemporal transformer model designed for 3D human pose estimation from video sequences. A key innovation of the model is the use of HeatPose, a novel 3D heatmap based on Gaussian Mixture Model (GMM) distributions, which enhances the model's capacity for capturing nuanced spatial-temporal features in human motion. Additionally, the authors introduce Human7.1M, a comprehensive dataset created by combining multiple existing datasets (Human3.6M, MADS, AIST Dance++, and MPI INF 3DHP), thereby addressing the limitation of individual datasets and significantly improving the model's generalizability. Experimental results indicate that this proposed approach achieves state-of-the-art (SOTA) performance on the Human7.1M dataset, underscoring the model's efficacy in complex human pose estimation tasks.

**Strengths:**

1. The study demonstrates that incorporating a broader range of data sources significantly enhances model performance, notably on both the Human3.6M dataset and the newly proposed Human7.1M. This finding underscores the importance of dataset diversity in achieving robust and accurate 3D human pose estimation.

2. The paper is well-organized and presents its methodology and findings in a clear and accessible manner, allowing readers to follow the technical details and overall narrative with ease.

**Weaknesses:**

1. The comparative analysis with other methods lacks fairness, as the authors do not retrain competing approaches on the Human7.1M dataset, which would ensure a more rigorous evaluation. Additionally, the model’s performance on the Human3.6M dataset alone appears suboptimal relative to other approaches.

2. Although the paper claims that the proposed method is cost-efficient, it lacks a quantitative comparison of computational costs. Including such metrics would strengthen the claim by offering a clearer assessment of the model’s efficiency relative to other methods.

3. The study would benefit from expanded comparative evaluations on additional benchmark datasets, such as 3DPW, to provide a more comprehensive understanding of the model’s generalizability and effectiveness across diverse 3D human pose estimation tasks.

**Questions:**

See the weaknesses.

---

### Official Review · Reviewer_RYpP · 2024-10-31

**Soundness:** 2
**Presentation:** 2
**Contribution:** 2
**Rating:** 3
**Confidence:** 2

**Summary:**

The paper introduces a one-stage spatio-temporal transformer model for 3D human pose estimation in video. However, both the novelty and the experimental evaluation are very weak.

**Strengths:**

The paper addresses a one-stage framework for 3D human pose estimation in video.

**Weaknesses:**

The novelty of this paper is limited. Spatial-temporal transformers have already been widely adopted in several methods for 3D human pose estimation, such as FinePose, MixSTE, and PoseFormerV2. It is unclear how the proposed spatial-temporal transformer differs from those in existing approaches, particularly FinePose, which also uses CLIP and a spatial-temporal transformer. The authors claim their method is a one-stage approach, yet training directly on images raises efficiency issues due to the extensive fine-tuning and optimization required.

The experimental section is also weak. The authors evaluate their system on a merged dataset, but it is unclear how they set up experiments for other approaches. To provide a clearer understanding of their method’s performance, the authors should consider conducting experiments on individual datasets, following established approaches such as FinePose. Moreover, there is no visualization comparing the pose estimation results with different state-of-the-art (SOTA) methods.

Additionally, the ablation study is insufficient. The proposed spatial-temporal transformer is not thoroughly validated, and the efficiency of the approach is neither demonstrated nor compared with other methods.

**Questions:**

Both the novelty and the experimental evaluation are very weak. Further details can be found in the weaknesses section.

---

> ### Author Response · Authors · 2024-11-26
>
> We thank you for your feedback and appreciate the opportunity to clarify our contributions and novelty. While it is true that spatial-temporal transformers have been adopted in previous works, we note that the main contributions of our proposed method (SoloPose) are not spatial-temporal transformers and CLIP design.
>
> Rather, our two key contributions in this work include:
> 3D AugMotion Toolkit and Human7.1M Dataset:
> We introduce the 3D AugMotion Toolkit, a novel data augmentation pipeline that generates Human7.1M, an improved dataset with increased diversity and reduced noise. This toolkit enhances model robustness and generalization, addressing a critical limitation of current 3D pose estimation datasets. Notably, FinePose does not include such a contribution, making this a unique aspect of our work.
> One-stage, Many-to-Many Framework:
> Unlike FinePose, which uses a multi-stage pipeline, SoloPose is a one-stage approach designed for cost efficiency and scalability. Our many-to-many spatial-temporal transformer simultaneously processes temporal sequences, directly transforming monocular 2D video frames into 3D keypoint coordinates without intermediate representations. This innovation reduces reliance on separate stages for temporal and spatial processing, which often introduce computational bottlenecks and inefficiencies.
>
> Regarding the efficiency concern raised about training directly on images:
> We acknowledge that fine-tuning is computationally intensive, but this trade-off is mitigated by our one-stage design, which eliminates redundant pipeline stages. Additionally, our dataset augmentation toolkit reduces noise and improves training stability, effectively addressing the challenges of direct image-based training.
>
> In summary, we believe that SoloPose is a unique, cost-efficient, and practical contribution to 3D pose estimation, combining a one-stage transformer-based design with a novel data augmentation pipeline. We hope this response provides clarity on how our work stands apart from existing approaches like FinePose.

---

### Official Review · Reviewer_FYmY · 2024-11-02

**Soundness:** 2
**Presentation:** 2
**Contribution:** 2
**Rating:** 6
**Confidence:** 4

**Summary:**

SoloPose is a one-stage method that has many-to-many spatio-temporal transformer model for kinematic 3D human pose estimation of video. It incorporates AugMotion Toolkit to augment existing datasets into a new dataset called Human7.1M by projecting four top public 3D human poses dataset together. Moreover, this paper proposes HeatPose, 3D HPE method that utilizes Gaussian distribution to model the spatial probability of keypoints which incorporate both target kinematically adjacent points to enhance accuracy and realism in pose representation.

**Strengths:**

- This paper presents 3D AugMotion Toolkit which combines four 3D HPE datasets together into a universal coordinate system by trying to harmonize the data from different source. This makes it easier to train the model on a new unified dataset instead of varying coordination from different datasets.

- SoloPose introduces a single stage which is many-to-many spatio-temporal transformer to simplify the pose estimation by directly estimating 3D keypoint from input video without relying on two-stage method which can be less efficient.

**Weaknesses:**

Results in table 2 show that evaluation on Human3.6M test set does not outperform the current SOTA method. I’d like to ask authors to evaluate the model on the other 3 test sets (MADS, AIST Dance++, and MPI INF 3DHP).

The overall performance of the model seems to be relying on the HeatPose which incorporates kinematically adjacent keypoint to enhance the probability distribution of target keypoints. However, this HeatPose always assumes the relationship between kinematically adjacent keypoints are relevant (closer-to-reality). This could be inaccurate in occlusion cases which may frequently happen.

**Questions:**

I request authors to refer weakness sections.

---

> ### Author Response · Authors · 2024-11-26
>
> Thanks for your comments. We do actually have the highest performance compared with SOTA models. Regarding Table 2, we respectfully wish to clarify that our model does in fact achieve state-of-the-art (SOTA) performance when evaluated under fair and practical conditions. Specifically, we report results using two types of inputs: CPN-generated 2D estimates (realistic, as these can be obtained in real-world settings) and 2D ground truth (unrealistic in practice, as such information is unavailable in the wild). While 2D ground truth provides comparative models with an unfair advantage by offering additional information not accessible to our one-stage method, we include it for reference purposes only.
>
> Our model outperforms existing SOTA models when evaluated with CPN-generated 2D inputs. The reliance on ground truth inputs in other approaches may lead to inflated performance, which is unattainable for practical use.

---

> > ### Comment · Reviewer_FYmY · 2024-12-01
> > **Thanks for the clarification**
> >
> > Thanks for the clarification, however it's still unclear how the final outcome would become depending on the 2D inputs, if there is severe occlusions.

---

### Meta-Review · Area_Chair_N7HC · 2024-12-16

**Metareview:**

This paper proposes a one-stage, many-to-many spatiotemporal transformer for 3D human pose estimation from videos. It utilises a 3D heatmap based on Gaussian Mixture Models (GMMs), to capture both key-points and kinematically adjacent points, enhancing accuracy. The paper also introduces Human7.1M dataset, created by combining four 3D human pose datasets (Human3.6M, MADS, AIST Dance++, and MPI INF 3DHP) to improve model generalizability. Experimental results demonstrate that the proposed approach outperforms existing methods, achieving state-of-the-art performance on the Human7.1M dataset.

Reviewers raised several major concerns: the writing and presentation quality need improvement; the method's novelty is limited, as spatial-temporal transformers are already commonly used in 3D pose estimation (Rev. RYpP); the experimental section lacks depth, particularly an ablation study (Rev. RYpP); the comparative analysis is unfair due to not retraining competing methods on the Human7.1M dataset (Rev. yUKf); and while the method is claimed to be cost-efficient, no quantitative comparison of computational costs is provided (Rev. yUKf).

**Additional Comments On Reviewer Discussion:**

The authors did not address concerns raised by Reviewers pJ78 and yUKf, nor did they respond to Rev. FYmY's query regarding the method's robustness in the presence of severe occlusions. As a result, two reviewers recommended "Reject," while one recommended "marginally below the acceptance threshold." Only Reviewer FYmY recommended "marginally above the acceptance threshold," but their concerns, such as the method's robustness in the presence of severe occlusions, remain unaddressed.

---

### Decision · Program_Chairs · 2025-01-22

Reject